# Optimization of Pin Position and Angle for Z-Pin-Reinforced Foam Core Sandwich Structures

**DOI:** 10.3390/ma16010352

**Published:** 2022-12-30

**Authors:** Eduardo Fischer Kerche, Agnė Kairytė, Sylwia Członka, Amanda Albertin Xavier da Silva, Maikson Luiz Passaia Tonatto, Francisco Luiz Bresolin, Rafael de Avila Delucis, Sandro Campos Amico

**Affiliations:** 1Post Graduate Program of Mining, Metallurgical and Materials Engineering (PPGE3M), Materials Department, Engineering School, Federal University of Rio Grande do Sul—UFRGS, Av. Bento Gonçalves 9500, Porto Alegre 91501-970, Brazil; 2Ford Motor Company, Instituto Euvaldo Lodi, Camaçari 42810-225, Brazil; 3Laboratory of Thermal Insulating Materials and Acoustics, Institute of Building Materials, Faculty of Civil Engineering, Vilnius Gediminas Technical University, Linkmenu St. 28, LT-08217 Vilnius, Lithuania; 4Institute of Polymer & Dye Technology, Lodz University of Technology, 90-924 Lodz, Poland; 5Post-Graduate Program in Mechanical Engineering, Federal University of Santa Maria, Cachoeira do Sul 96503-205, Brazil; 6Post-Graduate Program in Materials Science and Engineering, Federal University of Pelotas, Pelotas 96010-610, Brazil

**Keywords:** sandwich panels, three-point bending test, vacuum bag, finite element analysis, structure optimization

## Abstract

Sandwich panels (SP) are very promising components for structures as they ally high levels of specific stiffness and strength. Civil, marine and automotive industries are some examples of the sectors that use SPs frequently. This work demonstrates the potential of manufacturing Z-pin-reinforced foam core SPs, using a design strategy that indicated optimal values for both pin position and angle, keeping the same pin diameter as determined in a previous study. A simple search algorithm was applied to optimize each design, ensuring maximum flexural stiffness. Designs using optimal pin position, optimal pin angle and optimal values for both parameters are herein investigated using numerical and experimental approaches. The optimal pin position yielded an increase in flexural stiffness of around 8.0% when compared to the non-optimized design. In this same comparison, the optimal pin angle by itself increased the flexural stiffness by about 63.0%. Besides, the highest increase in the maximum load was found for those composites, molded with optimized levels of pin position and pin angle, which synergistically contributed to this result. All results were demonstrated with numerical and experimental results and there was a good agreement between them.

## 1. Introduction

Sandwich panels (SP) are very promising components for structures since they ally high levels of specific stiffness and strength, as well as thermal and acoustic insulation. Due to their outstanding performance, these structures are used in applications where both high mechanical properties and low weight are desirable, such as aerospace, marine and automotive industries [1,2]. In these applications, the designed loading conditions must be accurately predictable, and so SP development is moving towards the design of specific configurations to ensure that minimized levels of cost and weight accompany maximized performance [3].

SPs are usually composed of strong face-sheets, intermediated by a thick and lightweight core that has lower mechanical properties compared to the face. However, the overall structural efficiency of these parts depends on the properties of both faces and core, as well as the core-to-face bonding. For this reason, several manufacturing strategies have been investigated, including orthogonal weaving, stitching, tufting, and Z-pinning, which aim to reinforce the core and prevent undesired cohesive and interfacial failures [4,5,6,7,8]. These strategies have been inducing improvements in several core-related properties, especially flatwise flexural, flatwise compressive and shear properties.

Specifically, it was proved that pins inserted in the Z-direction (perpendicular to the SP length) may connect the faces, increasing the stress transferred between them and preventing core-skin delamination failures. This approach has shown improvements in three-point bending [3], flatwise compression [3,4] and tensile tests [5]. In this context, it is well known that the core, reinforced with regularly spaced Z-pins, may provide a limited improvement to the structural performance of SP. For instance, Delucis et al. [3] reinforced a poly (ethylene terephthalate) foam core by the use of transverse polymeric pins. Improvements in both flatwise compressive and flexural strength were reported for the SP. Furthermore, pins with a diameter below 3 mm led to failure by local SP buckling, while diameters larger than 4 mm showed manufacturing issues related to the incomplete filling of pins, when SP were manufactured by light resin transfer molding.

Indeed, many manufacturing techniques may be employed for SP, each aiming to reach a maximum performance with the lowest possible weight. Among them, vacuum infusion and light-resin transfer molding (L-RTM) are the most common molding processes used to manufacture sandwich panels [2,9,10]. In these cases, polymeric pins can be in situ produced during the panel manufacturing process, applying a pre-drilled core [11]. Then, the resin may infiltrate into the hole, filling it. However, as aforementioned, the incomplete filling of the pins’ cavities for certain pin diameters may consequently yield losses of mechanical properties [3]. Then, due to the non-influence of the Z-permeability of reinforcements, vacuum bag processing may help to prevent this manufacturing issue since this processing has been used for composite plates with high fiber volume fraction (V_f_), low density and good surface finishing [12].

The use of reinforcements at inclined angles may improve sandwich structural efficiency even more. This is due to the higher stiffness of structures reinforced with pins compared with those with a foam core support [13]. However, according to Guiqiong and Long [13], a shallow range of Z-pin angles is unlikely to significantly change the through-thickness properties, compared to all the pins being perfectly aligned in the orthogonal direction. Moreover, the collapse of a Z-pinned foam core is due to the buckling of the pin, and pin buckling is significantly dependent on the location of indenter. Then, pins with greater angle in relation to the horizontal plane are required to retain the overall sheer forces more, something usually supported by the SP’s core.

Usually, flexural tests generate the most realistic loads for SP since stresses are applied in the entire panel. For this reason, several experimental and numerical studies dealt with this test configuration to evaluate improvements in stiffness of both core and core-to-face bonding, as well as to understand damage and failure mechanisms [14,15,16]. For instance, Liu et al. [17] studied sandwich beams with pin-reinforced foam cores, which were modeled by FEM, using brick elements. Similar numerical approaches were proposed by [3,18,19]. In this context, there are several plate and shell theories which were developed specifically for sandwich structures [20]. However, these theories are not suitable for pin-reinforced cores, due to localized strains near to the pins.

Despite the studies regarding the manufacturing of enhanced Z-pinned SP, to the best of our knowledge there are no studies in the literature that investigate how both Z-pin position and angle may affect the structural performance of such SPs. Then, this study aims to present a simple algorithm to optimize these issues for an enhanced SP produced by a vacuum bag. For that, a mathematical approach was used to define the best values for pin position and pin angle. Numerical and experimental results were compared to fully explain the mechanical behavior of the structural panels.

## 2. Materials and Methods 

### 2.1. Optimization Process for Defining Pin Position and Pin Angle

Five different SP configurations were designed and manufactured. These configurations and respective nomenclatures are indicated in Table 1. The pin diameter was kept at 3.2 mm for all cases, as recommended by Delucis et al. [11].

For the optimization of both pin position and angle, a simple search algorithm (*fminsearch*) [21] was implemented using MATLAB R2020a, version 9.9.0.123456, ensuring a maximized force response on the bottom supports and considering two different study cases:Optimization of the position of orthogonal Z-pins, which was implemented for 0HD and 0OPT.Optimization of both the pin position and pin angle, keeping only one angle for all pins to facilitate the core drilling, which was implemented for AHD and AOPT.

The automated routine used for the optimization process was described in Figure 1. The initial design variables were inputted to the algorithm in order to obtain the representative values needed to execute the optimization process. These variables were qualified according to how they could improve the optimal design if selected on the basis of a finite element model (FEM), wherein the maximum force was returned as an objective function. The FE model again qualified the new interaction, and this process was repeated until the maximum criterion was reached. The optimization ended when the change in the value of the objective function was lower than 0.1 or the size of a step was lower than 0.25.

### 2.2. Numerical Flexural Tests for the Sandwich Structures

An FE model of the Z-pin-reinforced SP, subjected to three-point bending tests, has been developed. Pins’ diameter (d_p_), pins’ center spacing (s), angle orientation (θ), panel length (L) and panel width (W) were the primary structural parameters for defining the panels’ design. These parameters are typical for Z-pin-reinforced sandwich panels, as shows in Figure 2a,b. Figure 2b depicts the geometry of the 2D-dimensional SP, where the core thickness (t_c_) and the skin thickness (t_s_) are displayed.

Figure 2c shows the FEM used to simulate the three-point bending, where one quarter of symmetry is considered along the *x* axis and *z* axis. The bottom face nodes provide boundary conditions (pinned constraint) with were used to impose the constraints of the bending test supports. A displacement equal to 10 mm is applied in upper face nodes to represent the displacement applied by indenter. As the FEM is considered using a quarter of symmetry, symmetric constraints are imposed on the surface, as indicated in Figure 2c.

The finite element model uses S4 linear shell elements for the GFRP skins and C3D8 eight-node linear elements for the polyester pins and PET foam. Mesh convergence, related to the through-the-thickness elements, is analyzed based on the maximum load recorded in the three-point bending test, which converged for a mesh with 40,462 elements, as shown in Figure 2c. The foam is geometrically modeled, with holes used to assemble the pins. Then, the foam is separately meshed, and assembled with pins and faces that relate to the degrees of freedom of the pins to those of the foam.

Mechanical responses of the sandwich structures were determined using Abaqus^TM^/Standard v. 6.13 for the GFRP skins, whereas the polyester pins used a non-linear model. The static model is used to perform the three-point bending test. The skins are modelled using the elastic properties described in Table 2. In order to simulate the non-linear effects of the PET foam, a crushable isotropic hardening foam model is used. The calibration of the crushable foam’s parameters was performed in our previous study [3].

Table 2 shows the parameters used in this model. Besides, the hardening curve used in the model is built using the relation $\sigma_{y}^{\mathrm{pl}}=0.624+0.053\cdot e^{\frac{\varepsilon_{y}^{\mathrm{pl}}}{0.554}}$, where $\sigma_{y}^{\mathrm{pl}}$ is the plastic yield stress and $\varepsilon_{y}^{\mathrm{pl}}$ is the uniaxial plastic strain. The polyester pin was treated as an isotropic material, where the elastic modulus (E) and Poisson’s ratio (ν) were 1300 MPa and 0.34, respectively, obtained from experimental tensile tests, following ASTM D638–14, speed of test 5 mm/min. in a universal test machine (Instron^®^), equipped with a load cell of 5kN.

### 2.3. Optimization Problem

The optimization method consisted of increasing the flexural stiffness by the sandwich panels. The sandwich panel design (“D”) is a real-valued array of parameters that defines the structure’s geometry (Equation (1)). Based on the foam core configuration shown in Figure 3a,b, the design variables are pin center spacing (pin position), i.e., s_1_, s_2_, s_3_, s_4_ and s_5_, and pin angle orientation (θ). The other SP’s dimensions are kept constant during the optimization process.
(1)$$D=\{s_{1}{,\;s}_{2}{,\;s}_{3}{,\;s}_{4}{,s}_{5},\theta \}$$

Given a function of F_max_ (which describes the maximum reaction force found in nodes), the displacement was applied. The displacement is kept constant to obtain the maximum force in order to obtain the stiffness effect. Then, the following optimization problem is formulated:(2)$$\;\underset{D\in x}{\min}\{{\;-F}_{\max}\left(D\right)\}$$
where “x” is the known analysis domain, defined by design variables within the FE model.

Finally, a sensitivity study is conducted in order to evaluate the effect of design variables on the objective function. Thus, other simulations are carried out in the optimized designs 0OPT and AOPT, where infinitesimal variations are carried out in the design variable “x” in order to evaluate the effect on the variation of the flexural force “(F)”.

### 2.4. Manufacture of the Sandwich Structures

E-glass fiber mats (300 g/m² aerial density) and 12 mm thick commercial closed-cell PET–foam cores (Divinycell PN80) were purchased from Barracuda^®^. Unsaturated polyester resin (Alpha 163 from Embrapol/initiator (acetyl-acetone peroxide) system) was used to manufacture the sandwich panels using a vacuum bag (Figure 3). In the processing, a layer of release agent was applied on the delimited molding area. Then, like a manual lamination process, alternating layers of polyester resins and glass mats were applied on this smooth surface until a 5-layered glass-based laminate was reached.

After that, the drilled PET foam core (with holes’ positions and angles made by a special drilling machine) was positioned and a cover layer of resin was applied, filling the pins’ holes. This was followed by another five layers of glass mats alternated with resin layers, as aforementioned. Afterward, layers of peel ply, flow mesh and breather were positioned in this sequence. Vacuum channels and spiral tubes were positioned near to the injection area, and the vacuum bag was sealed with the aid of a vacuum bag film and tacky tape. The cavity was then evacuated (−100 kPa or −1 bar), removing air and compacting the reinforcements. The plates were cured for 24 h under the mentioned vacuum, and the composite was demolded and subjected to a post-curing process at 80 °C for 2 h, as determined by the resin supplier.

### 2.5. Tensile Tests for the Pins and Skins

In order to obtain the elastic properties of the polyester used to manufacture the pins, neat resin tensile tests’ samples were manufactured by casting using silicone rubber molds. Then, the tensile tests were performed, according to ASTM D638, using a cross-head speed of 5 mm/min. Other tensile tests were performed according to ASTM D3039, using five 5-layered glass mats/polyester laminates with the dimensions of 250 mm × 25 mm. The specimens were tested until rupturing at a cross-head speed of 2 mm/min. Longitudinal and transverse strains were recorded using clip gauges, aiming to obtain the face-sheets’ mechanical properties, essential for the FEM.

### 2.6. Experimental Flexural Tests for the Sandwich Structures

Flexural tests followed the standard ASTM C393. Six specimens (dimensions: 200 mm × 75 mm) per sandwich panel configuration were loaded at a speed rate of 1.0 mm/min (up to failure) and a span/thickness ratio of 32 (≈128 mm). SP’s inertia moment (I), flexural stiffness (K), elastic modulus (E_f_) and core shear ultimate strength (F_s_^ult^) were calculated according to Equations (3)–(6), respectively.
(3)$$I=\frac{w{\cdot t}_{1\mathrm{fs}}}{2}\left[\frac{t_{1\mathrm{fs}}{}^{2}}{3}+{\left({c+t}_{1\mathrm{fs}}\right)}^{2}\right]$$
(4)$$K=\frac{P_{\max}}{\Delta }$$
(5)$$E_{f}=\frac{S^{3}}{48\cdot I}\times K$$
(6)$$F_{s}^{\mathrm{ult}}=\frac{P_{\max}}{\left(t_{1\mathrm{fs}}+c\right)\cdot w}$$
where: w is the specimens’ width, t_1fs_ is the thickness of a single face sheet, c is the core thickness, P_max_ is maximum applied load at the specimen’s middle spam, ∆ is the machine displacement at maximum load, S is the test spam, and e is the SP’s thickness.

Inferential statistical analysis was performed on the obtained mechanical properties and apparent density. The normality and homogeneity of each level (type of pin configuration) were verified by Shapiro–Wilk and Levane tests, respectively. After that, one-way ANOVA and averaging tests were performed, following the LSD Fischer procedure with 95% confidence. The statistical analysis was performed using Static Graphics^®^ software, version 16.1.03 (Spanish version).

## 3. Results and Discussion

### 3.1. Optimization Process for Defining Pin Position and Pin Angle

Figure 4 shows the convergence of the optimization process for the AOPT design, wherein the values of the objective function for the 1–250 interaction range are shown. The values are dispersed in the early interactions, which indicates that the algorithm poorly designs the function for optimized objective values. Then, more interactions are required. The values stabilize above 150 interactions, and this indicates a satisfactory optimization process. Besides, all other panels showed similar convergence curves.

Table 3 lists the designed variables, corresponding objective functions and flexural modulus values for the chosen designs (see Equation (3) and Figure 2). Design 0HD presents a flexural modulus 7.7% higher than the reference design (R). This explains that the insertion of pins, without optimizing the positioning and orientation angle, provides an increase in maximum force and, consequently, an increase in flexural modulus [17]. Design 0OPT exhibits similar flexural modulus (0.7% higher), when compared to the non-optimized design (0HD). Design AHD has about the same position of the pins as 0HD and the angle orientation has been optimized, revealing 12.3% increases in flexural modulus when compared to 0HD. Finally, the optimization process of the design AOPT demonstrated higher maximum load and flexural modulus, when compared with the other designs.

The maximum principal stress field of the optimal designs are shown in Figure 5. This stress is important as it gives an idea of how each pin is contributing to the overall stiffness of the whole SP. Design AOPT (the design with higher elastic modulus) features the maximum principal stress, with a larger area than other designs. Design 0OPT exhibited a similar maximum principal stress than the non-optimized design (0HD) despite differences in pin positions between the two designs. Designs AHD and AOPT presented similar maximum principal stress distributions and the pins have an orientation angle that tended to maximize the SP stiffness.

### 3.2. Experimental Flexural Tests for the Sandwich Structures

Regarding the experimental results, Figure 6 presents the median curves for the modified SPs, submitted to the 3-point bending tests. Table 4 presents the main results for the experimental data. The panel with the pin distances, optimized and oriented at 0° (0OPT), presented the higher modulus of elasticity, while those with the optimized distance of angled pins (i.e., those AOPT) showed the lower results. Nonetheless, the later presented equal values of E_f_ to the reference sample (i.e., those R). From these results, it’s possible to see that the pins’ position optimization improved the structure flexural stiffness ≈ 70%, with a small reduction (15%) in the flexural strength, compared to the reference sandwich (R).

However, as shown in Figure 6, the reference sample (R) exhibited a greater capacity for bending strain to failure (flexural strain) compared to the sandwiches that exhibited higher stiffness values (i.e., those 0HD, 0OPT and AHD). This behavior is explained by the stress concentration, generated by stiffness’ differences between the resin pins, and the PET foam core (different deformations are generated for the same applied load) [22]. Finally, those samples AOPT presented the highest deformation capability, without suffering an abrupt break during failure (up to 7% flexural deformation), which was perhaps related to the pins’ orientation (i.e., on the core shear stresses load) [23].

It is important to stress that despite the improvements on sandwich modulus, the pins acted as stress concentrators in the core, explaining the lower results for flexural strength (Table 4). The reference sandwich, R, presented the highest F_S_, compared to all the other samples, and these presented equal values for F_S_, showing that the optimization or orientation does not interfere on the core stress concentration. 

FEM used in the optimization process does not take into account the failure of any of the sandwich components, explaining the differences found between experimental and numerical results (shown in Table 3). Indeed, there were significant differences for all samples, when comparing FEM and the experimental tests. For instance, E_f_ reached differences ≈10, 14, 54, 09 and 51% for those R, 0HD, 0OPT, AHD and AOPT, respectively, comparing the values from Table 3 to those from Table 4.

For the apparent density (Table 4), an unexpectedly significant reduction is reported for those AOPT and AHD, compared to those panels with non-angled pins (i.e., those 0HD and 0OPT) which display no significant difference compared to the reference panel (R). This behavior may be related to the non-fulfilling of the pins when the sandwich is manufactured, especially when pins present an angle [3]. Despite these significant differences, a significant increment for those 0HD and 0OPT is reported compared to R. This may be related to the higher pin filling due to the 0° angle in relation to the horizontal plan, which can facilitate the resin penetration into the holes.

Regarding specific strength (Table 4), there was a relevant decrease for the samples with the pins, regardless of its angled orientation. This result may be related to the higher weight of the overall structure when the pins are used in the foam. 

Figure 7 presents the samples’ failure patterns of the Z-pin-reinforced sandwich panels. 0HD and 0OPT presented a core failure with similar patterns, an occurrence which highlights the non-significance of the optimization on the core stress concentration. From Figure 6, those R and AOPT samples have the greater bending strain. On the other hand, the failure patterns are different. The reference sample (R) shows core failure close to the top face sheet, where the core is under tensile conditions. The AOPT sample exhibited core compression failure, compressing the core foam cells and not suffering an abrupt load break and failure, as presented in Figure 6. Finally, those AHD panels presented a failure similar to that displayed by 0HD and 0OPT, perhaps due to the similar stiffness of their modified core.

Figure 8 presents images of the cross section of some samples after mechanical testing. It is possible to see that both samples presented a better filling of holes by the resin compared with a previous study, where the sandwich panels were manufactured by L-RTM, instead vacuum bag process [3]. On the other hand, those pins with an angled orientation through the thickness (Figure 8b) presented apparent poorly resin filling capacity. In this case, the angle may difficult as the resin infiltration and distribution throw the hole, which may reflect on the overall sandwich panels’ response, as also discussed above.

### 3.3. Numerical Flexural Tests for the Sandwich Structures

In order to observe the mechanical behavior improvement obtained in cases I and II, the optimized beams are compared with their non-optimized counterparts. Numerical and experimental results’ force response is presented in Figure 9. In general, the numerical model presents a good agreement in flexural stiffness when compared with experimental results. The non-linear effect evidenced in the numerical response is due to the crushable foam model considered for the foam. Then, the region after the yield of the foam is evidenced. In other words, this model produced a bi-linear effect, with the change in stiffness in the intermediate region of the curves (flexural load ≈ 1000 N) taking place in the foam flow region in which it passes from linear to non-linear behavior. However, in several experimental curves (mainly 0HD, 0OPT and AOPT), changes in stiffness in the same load region are observed, showing similarity to the bilinear effect found in the numerical model.

The AOPT design presents the highest difference in flexural stiffness between numerical model and experimental results, when compared with other designs. This difference may be attributed to the difficulty the model experienced in predicting the deformation field in the indentation region. The core indentation is well defined for this design which is closed of the indenter for 3-point bending test, as shown in Figure 7, while in numerical model this effect is not captured by the crushable foam model. Another factor that may cause this difference is associated with the difficulty of manufacturing the inclined pins, since the difference in stiffness is also identified in the 0OPT design. Furthermore, it is important to mention that failure models are not implemented in the simulations, since the objective is to optimize flexural stiffness. The abrupt drops in the experimental curves may be associated with failures of the core, face, pins or interface between these constituents. 

Finally, the SP with different configurations presented different values for maximum strain, as shown in Figure 9, especially for the 0HD and 0OPT designs, compared with the reference panel (R). These results may be related to the premature failure of the pins, caused by unbalance of load during the test [3]. Despite the differences, the lower displacement may be interesting when the SP is destined for floors and roofs, when a structure with low deformability and with stiffness is required.

Figure 10 shows the sensitivity analysis results for 0OPT and AOPT designs. The largest changes in the flexural force variation, with respect to the geometric parameter variation $\left(\frac{\partial F}{\partial x}\right)$, occurs for the displacement (δ) equal to 2 mm for 0OPT design (see Figure 9a). This effect is attributed to initial yield of the core, also observed in Figure 7. In this displacement, the flexural load presents higher changes for the design variable s_2_, s_3_ and s_4_. This effect has been attributed to the fact that these parameters are closer to the central region between the indenter and support (see Figure 2), which presents higher shear stresses in the core. Besides, in all ranges of displacement, sensitivity functions have negative values. This means that flexural force “F” decreases when the geometric parameters (s_1_, s_2_, s_3_, s_4_, s_5_) increase.

In high displacements, the flexural force has been shown to be more sensitive to the s_3_ dimension, while the other dimensions have similar behavior. Regarding AOPT design (see Figure 10b), a similar behavior is observed where greater changes in the flexural force occur, with displacement equal to 2 mm. In this case, it is also noted that the angle of the pins (θ) has higher values of $\left(\frac{\partial F}{\partial x}\right)$, which means that the variation in this design variable also significantly modifies the flexural force. Finally, s_5_ has shown that sensitivity functions have positive values for high displacements. This means that an increase in the flexural force occurs with the increase in this design variable. This variable represents the position of the pins close to the indenter, which favors the reduction in indentation formation.

## 4. Conclusions

An objective optimization of Z-pin in situ which generated reinforcements for enhanced sandwich structures, under a bending load, has been proposed. The optimization has been performed using FEA models, using a custom routine integrated with a simple search algorithm, to analyze the designs. A maximization of the maximum flexural load was used as an objective function, aiming to increase the flexural stiffness of SPs. The optimized designs were obtained using four different configurations. FEA models were used to simulate three-point bending tests, and those SPs with optimized pins’ positions (0OPT) presented an increase ≈ 8.0% in flexural modulus compared to the non-optimized design (0HD). Furthermore, the optimization of the pin angle (AHD) increased the flexural modulus ≈ 63.0% in comparison with 0HD, and the most significant increase in maximum stiffness occurs when the pins’ position and angle are simultaneously optimized (AOPT). 

Despite the improvement in mechanical response, when SP was subjected to simulation of three-point bending, angled pins cause difficulties in filling the pins during the manufacturing of the SPs. Then, manufacturing issues may be the main issue responsible for the non-significant improvement in both F_s_^ult^ and flexural modulus of the angled pins, although improvement in such properties occurred for the non-angled pins’ structures compared to the reference sample. Finally, in general, the optimized designs presented good agreement with experimental results. This study presents for the first time an optimization of pins, generated in situ for the improvement of the flexural properties of SPs. The developed structures may be used in applications where a high specific strength and stiffness are required.

## Figures and Tables

**Figure 1 materials-16-00352-f001:**
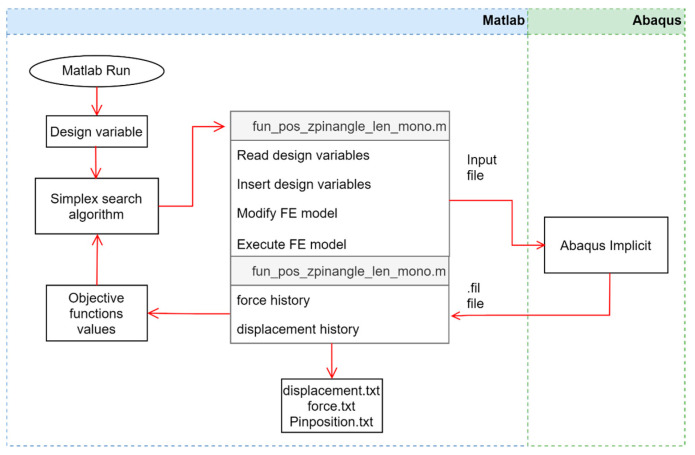
Scheme for the optimization routine with simplex search algorithm in Matlab^®^ and ABAQUS^®^.

**Figure 2 materials-16-00352-f002:**
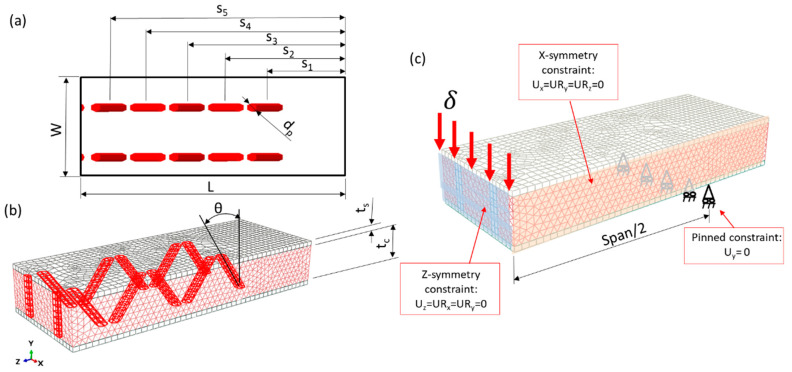
(**a**) A quarter of the section for the dimensions of the Z-pin-reinforced sandwich panels used for the modeling, with the respective design variables (s) (i.e., pin center spacing in relation to the SP center), half of length (L) and width (W), (**b**) 3-D panel geometry with its respective pin angle (θ) and (**c**) finite element model of the sandwich panel for the three-point bending test simulation.

**Figure 3 materials-16-00352-f003:**
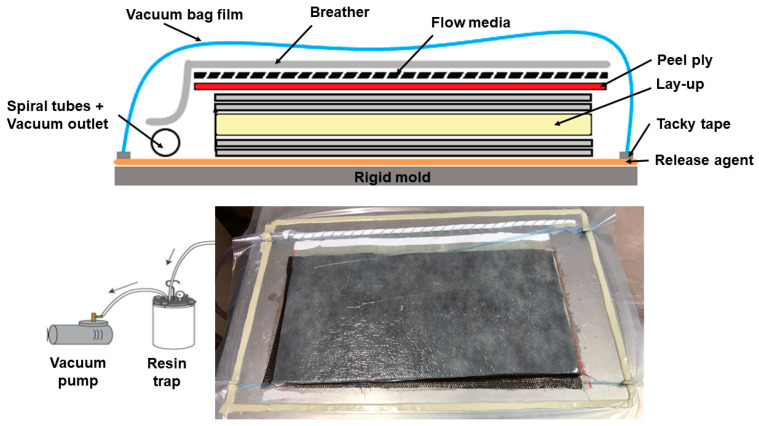
Vacuum bag processing, for the SP manufacturing.

**Figure 4 materials-16-00352-f004:**
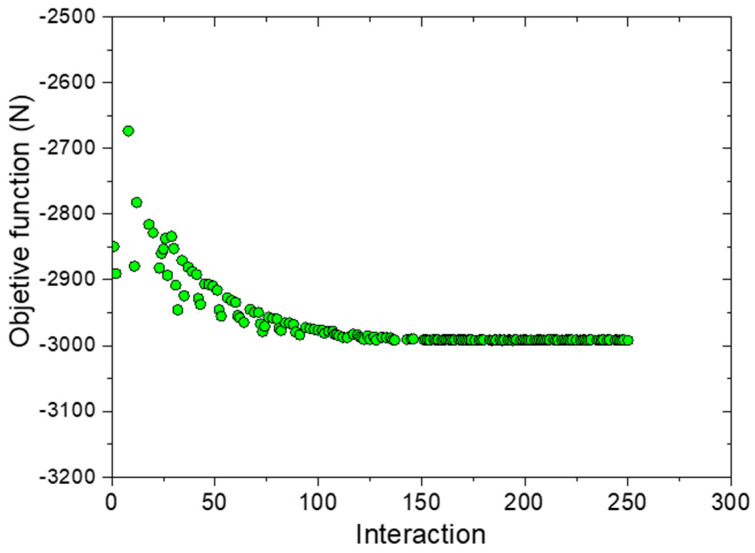
Convergence curve of the optimization process for the AOPT panel.

**Figure 5 materials-16-00352-f005:**
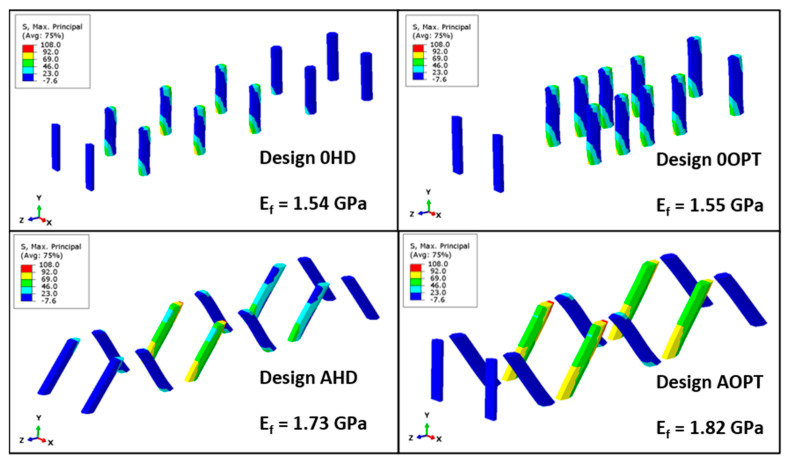
Maximum principal stress in pins of selected designs on the simplex optimization process for displacement of 10 mm.

**Figure 6 materials-16-00352-f006:**
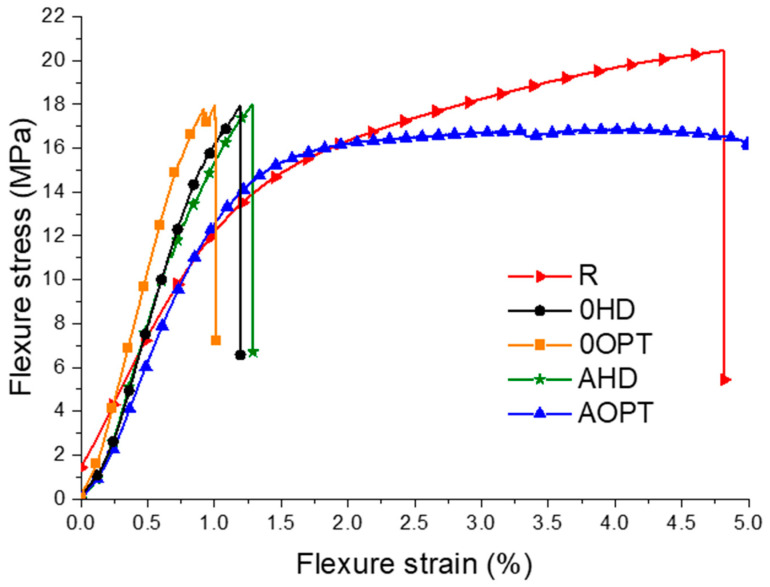
Flexure stress–strain of all sandwich’s structures.

**Figure 7 materials-16-00352-f007:**
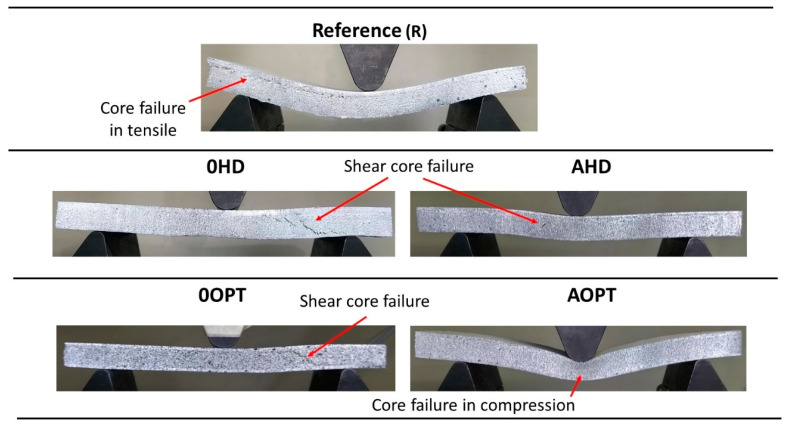
Sandwich panels’ failure patterns.

**Figure 8 materials-16-00352-f008:**
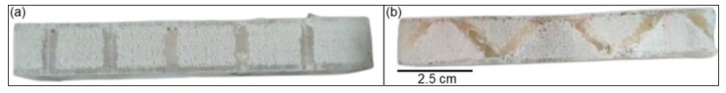
Sandwich panels’ cross section (**a**) 0HD and (**b**) AHD.

**Figure 9 materials-16-00352-f009:**
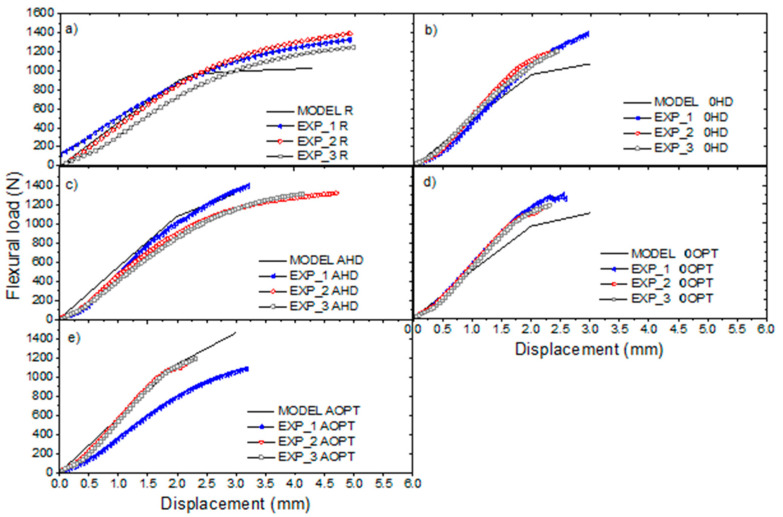
Numerical and experimental flexural load vs. displacement curves for (**a**) R, (**b**) 0HD, (**c**) AHD, (**d**) 0OPT and (**e**) AOPT designs. The EXP_X curves are referenced as the experimental curve for each sample (three for each SP configuration).

**Figure 10 materials-16-00352-f010:**
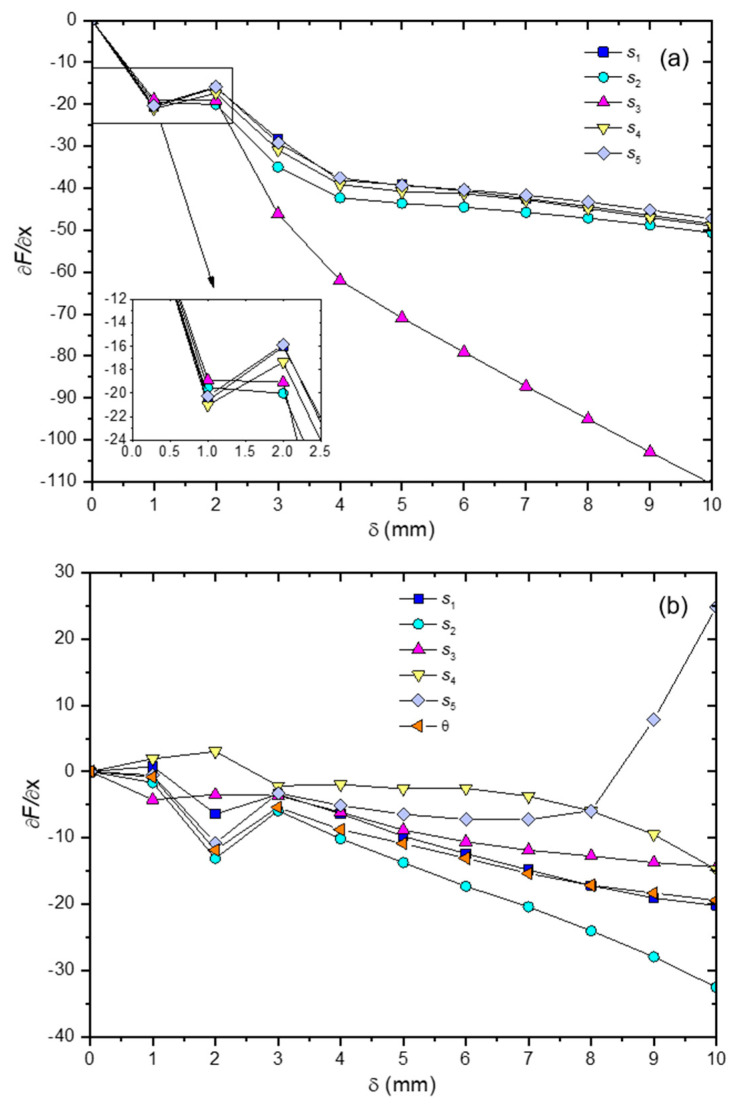
Sensitivity analysis results of flexural force variation with respect to the geometric parameters variation $\left(\frac{\partial F}{\partial x}\right)$ in function of displacement (δ) for the (**a**) 0OPT and (**b**) AOPT designs.

**Table 1 materials-16-00352-t001:** Z-pin-reinforced sandwich panels’ nomenclature and configuration.

Configuration	Nomenclature	Pin Diameter (mm)
No pins—reference	R	3.2
Pins oriented 0° throw the thickness and homogeneous distributed	0HD	
Pins oriented 0° throw the thickness and the distance between pins optimized	0OPT	
Pins with optimized angle orientation throw the thickness and homogeneously distributed	AHD	
Pins with optimized angle orientation and distance throw the thickness	AOPT	

**Table 2 materials-16-00352-t002:** Material’s properties of the GFRP skins and crushable foam’s parameters of the PET foams.

Material	Property	Symbol	Value
GFRP skin	Longitudinal elastic modulus *	E_11_ (MPa)	10,500
	Transverse elastic modulus *	E_22_ (MPa)	10,500
	Poisson’s ratio *	ν_12_	0.3
	In-plane shear modulus *	G_12_ (MPa)	4038
PET foam	Elastic modulus **	E (MPa)	60
	Poisson’s ratio **	ν_el_	0.35
	Compressive yield stress ratio **	K	1.6656
	Plastic Poisson’s ratio **	ν_pl_	0.04

* Based on mechanical tests. ** Based on [22,23].

**Table 3 materials-16-00352-t003:** Optimal designs found for each panel configuration.

Configuration	S_1_ *	S_2_ *	S_3_ *	S_4_ *	S_5_ *	θ (°)	$F_{\max}^{\delta =10\;\mathrm{mm}}\;$	E_f_ (GPa)
R	-	-	-	-	-	-	1.124	1.43
0HD	9.00	27.2	45.40	63.60	81.80	0	1.452	1.54
0OPT	35.75	51.00	59.65	66.30	73.98	0	1.566	1.55
AHD	9.00	27.20	45.40	63.60	81.80	35.5	2.364	1.73
AOPT	30.60	45.26	59.93	74.58	89.24	36.2	2.963	1.82

* S_x_ are the design variables (pin center spacing, or pin position, defined by Equation (1)).

**Table 4 materials-16-00352-t004:** Compilation of the sandwich structures’ properties from experimental tests.

Configuration	E_f_ (GPa) *	F_s_^ult^ (MPa) *	Apparent Density (kg/m^3^) *	Specific Strength (kPa·m^3^/kg^3^)
R	1.3 ± 0.11 ^A^	20.3 ± 0.47 ^B^	482.3 ± 19.5 ^A^	41.9 ± 2.4
0HD	1.8 ± 0.21 ^B^	18.2 ± 1.62 ^A^	542.4 ± 28.2 ^B^	33.5 ± 5.7
0OPT	2.4 ± 0.06 ^C^	17.2 ± 0.77 ^A^	540.8 ± 18.9 ^B^	31.8 ± 4.0
AHD	1.9 ± 0.18 ^B^	17.5 ± 0.61 ^A^	475.9 ± 15.3 ^A^	36.7 ± 3.9
AOPT	1.2 ± 0.14 ^A^	17.2 ± 1.04 ^A^	470.7 ± 21.0 ^A^	36.5 ± 4.9

* Different uppercase letters represent a statistical difference between the means, for each sample.

## Data Availability

Not applicable.

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
