# Peer review of "Optimization of Pin Position and Angle for Z-Pin-Reinforced Foam Core Sandwich Structures"

_materials, 2022, doi:10.3390/ma16010352_

Round 1

Reviewer 1 Report

1. The content needs to be corrected for many grammatical errors. 

2. The caption in Figure 2 is not clear. May be corrected. 

3. In figure 6, results for 0OPT section are not shown whereas the discussion comments on the same. 

4. As shown in figure 9(e), caption is not clear. Are the experimental comparisons done for different type of pin profiles. 

5. The simulation result in figure 9, show a bi-linear type of material behavior. The authors can clarify regarding the material model adopted in simulations and its suitability with experimental observations. 

Author Response

Reviewer 1:

We appreciate your interest by our work and the time dedicated for the deep revision of the manuscript. Following, you find the main questions related to suggested modifications of our work, as well as an answer for each issue. All suggested modifications, as well as English modifications are highlighted in yellow in the manuscript.

  1. The content needs to be corrected for many grammatical errors.

Comment: We performed an extensive revision of our manuscript, aiming to identify and correct grammatical errors and English spelling.

  1. The caption in Figure 2 is not clear. May be corrected.

Comment: We improved the Figure caption, as required.

  1. In figure 6, results for 0OPT section are not shown whereas the discussion comments on the same.

Comment: Thank you to pay attention on that. Actually, the samples’ name where wrong. Where you read 90HD and 90OPT are 0HD and 0OPT, respectively. We corrected the code in the image.

  1. As shown in figure 9(e), caption is not clear. Are the experimental comparisons done for different type of pin profiles.

Comment: Thank you again. The samples’ name where again wrong. Fig. 9(e) represents the curves for those SP with AOPT configuration. Furthermore, we included an additional comment in the figure caption, to better elucidate to the reader “The EXP_X curves are referenced as the experimental curve for the sample (three for each SP configuration)”.

  1. The simulation result in figure 9, show a bi-linear type of material behavior. The authors can clarify regarding the material model adopted in simulations and its suitability with experimental observations.

Comment: The bilinear models presented in all cases, arises from the non-linear effect caused by the Crushable foam model for the foam, as described in the methodology “Analysis of the sandwich structures was performed using AbaqusTM/Standard v. 6.13 for the GFRP skins, and the polyester pins used a non-linear model imposed due the foam’s nonlinearities”. We also included a sentence in the main text explaining the results “The non-linear effect evidenced in the numerical response is due to the Crushable foam model considered for the foam, then, the region after the yield of the foam is evidenced. In other words, this model produced a bi-linear effect, with the change in stiffness in the intermediate region of the curves (flexural load ≈1000 N) being the foam flow region in which it passes from linear to non-linear behavior. However, in several experimental curves (mainly 0HD, 0OPT and AOPT) changes in stiffness in the same load region are observed, which shows similarity with the bilinear effect found in the numerical model”.

Reviewer 2 Report

The manuscript titled 'Optimization of pin position and angle for Z-pin reinforced foam core sandwich structures' very impressive and the novelty of the study is good. However, some small modification has to be done for the further publication process.

1. Table 1 adds some gaps between every row.
2. Add comparison table for numerical and practical value
3. All pin orientation schematic figures have to add.
4. mention the application of the present study in the abstract and conclusion

Author Response

Reviewer 2:

The manuscript titled 'Optimization of pin position and angle for Z-pin reinforced foam core sandwich structures' very impressive and the novelty of the study is good. However, some small modification has to be done for the further publication process.

Comment: Thank you for your significant contribution in improving the quality of our work and deep review the manuscript. Above you find your comments and the response for each suggestion. All suggested modifications are highlighted in yellow in the manuscript.

  1. Table 1 adds some gaps between every row.

Comment: The gaps were added.

  1. Add comparison table for numerical and practical value.

Comment: Thank you for your opinion. The results from FEA models are presented in Table 3 and from Experimental test in Table 4. We separated the results in this format aiming to separate in sections the results from numerical and experimental parts. On the other hand, taking into account your consideration, we included a paragraph comparing the results from FEA to those from experimental section, highlighted before Table 4.

  1. All pin orientation schematic figures have to add.

Comment: All samples’ configurations are presented in Figure 2 and Fig. 5. Fig. 2 presents how the optimization problem was defined, using as input the pins position and/or pins’ angle. On the other hand, Fig. 5 presents the results of strains, when the SP structures are submitted to the 3-point bending test, also related to the numerical study.

  1. mention the application of the present study in the abstract and conclusion.

Comment: We included a sentence in abstract and conclusion regarding SP application.

Reviewer 3 Report

The manuscript studies the optimal design of a sandwich structure with a z-pin reinforced foam core. The study mainly focuses on two design variables of the core: spacing lengths between the pins and the pin's angle. The target design parameter is the maximum force in the flexural test. The study also provides experiments to validate the simulation results. The specimen manufacturing process is presented. Several concerns are risen by the reviewer and should be addressed well before recommending a publication for the manuscript. 

1. Several sandwich specimens are manufactured for experiments, their densities are different from each other, so it would be better to compare their strengths in terms of specific strength. 

2. It would be a similar question for the optimal designs when the specific strength should be considered.

3. In the simulation, did the authors consider the material property for the pin? is it polyester or foam? if so, please include one.

4. The reviewer is concerned whether the excessive stress existing in the core is considered in the design optimization. 

5. According to the numerical and experimental results in Fig. 9, most of the configurations have a similar value of flexural load but a much different value of flexural displacement, so it would be nice if the authors could give discussions on their engineering applications corresponding to the different behaviors of each design configuration.

6. Figure's quality shold be improved.

7. The conclusion is not clear and too lengthy.

Author Response

Reviewer 3

The manuscript studies the optimal design of a sandwich structure with a z-pin reinforced foam core. The study mainly focuses on two design variables of the core: spacing lengths between the pins and the pin's angle. The target design parameter is the maximum force in the flexural test. The study also provides experiments to validate the simulation results. The specimen manufacturing process is presented. Several concerns are risen by the reviewer and should be addressed well before recommending a publication for the manuscript. 

Comment: Thank you for your great contribution and interest in improving the manuscript quality. Your relevant contributions helped us to improve the quality of our manuscript. All suggested modifications are highlighted in yellow in the manuscript.

  1. Several sandwich specimens are manufactured for experiments, their densities are different from each other, so it would be better to compare their strengths in terms of specific strength. 

 Comment: We included a new column in Table 4 containing the results of specific strength for all samples.

  1. It would be a similar question for the optimal designs when the specific strength should be considered.

Comment: The Simulations were performed aiming to optimize the pins’ distribution and angle, aiming to improve the Flexural Stiffness, also related to the improvement in the Flexural MODULUS. For the prediction of maximum strength, many parameters need to be taken into account and it was not the focus of our study. Furthermore, it is so hard to predict the panels final density exactly, due to the irregular resin penetration into the hole, when the material is manufactured. For these reasons, specific strength was not considered for the simulations

  1. In the simulation, did the authors consider the material property for the pin? is it polyester or foam? if so, please include one.

 Comment: Yes, the polyester pin was treated as an isotropic material, where the elastic modulus (E) and Poisson’s ratio (ν) were 1300 MPa and 0.34, respectively, obtained from experimental tensile tests, following ASTM D638–14, speed of test 5 mm/min. in a universal test machine (Instron®), equipped with a load cell of 5kN. This information was included in the main text, before Table 2

  1. The reviewer is concerned whether the excessive stress existing in the core is considered in the design optimization. 

Comment: Yes, for the sandwich panels’ flexural properties simulations, the skins were modelled using the elastic properties described in Table 2. In order to simulate the non-linear effects of the PET foam, a crushable isotropic hardening foam model is used. The calibration of the crushable foam’s parameters is performed was our previous study [3] (doi:10.1177/1099636219830145).

  1. According to the numerical and experimental results in Fig. 9, most of the configurations have a similar value of flexural load but a much different value of flexural displacement, so it would be nice if the authors could give discussions on their engineering applications corresponding to the different behaviors of each design configuration.

Comment: We included a sentence after Fig. 9 regarding the discussion about differences in flexural strain. Moreover, we also included some comment about the different applications for these materials with different flexural strains.

  1. Figure's quality should be improved.

Comment: We improved the quality of some Figures.

  1. The conclusion is not clear and too lengthy.

Comment: We improved the conclusion quality and clarification.

Round 2

Reviewer 3 Report

The manuscript is recommended for publication.